# Direct Advantage Estimation for Scalable and Sample-efficient Deep Reinforcement Learning

## Abstract

Direct Advantage Estimation (DAE) has been shown to improve the sample efficiency of deep reinforcement learning. However, its reliance on full environment observability limits applicability in realistic settings. In the present work, we (i) extend DAE to partially observable domains with minimal modifications, and (ii) reduce its computational overhead by introducing discrete latent dynamics models to approximate transition probabilities efficiently. We evaluate our approach on the Arcade Learning Environment and find that DAE scales with function approximator capacity while maintaining high sample efficiency.

## 1 Introduction

Real-world decision-making problems often involve incomplete information, where observations received by the agents are not enough to fully determine the underlying state of the system. For example, a robot navigating a building may only have a local view of its surroundings; a doctor has to decide the course of treatment for a patient based on a limited set of test results. The Partially Observable Markov Decision Process (POMDP) framework (Kaelbling et al., 1998) provides a generalization of the fully observable MDP framework (Puterman, 2014) to tackle these problems.

While reinforcement learning (RL) (Sutton & Barto, 2018) paired with deep neural networks (deep RL) has achieved unprecedented results in various domains (Mnih et al., 2015; Berner et al., 2019; Schrittwieser et al., 2020; Ouyang et al., 2022; Wurman et al., 2022), it is known to be challenging to train and often requires millions or billions of samples (Henderson et al., 2018). Approximating the state(-action) value functions ($Q^\pi$ or $V^\pi$) is a crucial part of training deep RL agents. However, these functions are often highly non-stationary and difficult to learn due to their dependencies on the policy. Pan et al. (2022) demonstrated that the advantage function is more stable under policy variations, proposing Direct Advantage Estimation (DAE), a multi-step method, to learn the advantage function directly. DAE demonstrated strong empirical performance, but is restricted to on-policy settings. Later, Pan & Schölkopf (2024) observed that the return of a trajectory can be decomposed into two different advantage functions, which enabled a natural generalization of DAE to off-policy settings. Off-policy DAE was reported to further improve the sample efficiency of DAE; however, the method suffers from significantly increased computational complexity due to the need to learn a high dimensional generative model to approximate the transition probabilities.

The present work addresses the two limitations of Off-policy DAE: (1) applicability in partially observable domains, and (2) high computational overhead. More specifically, the contributions are:

- We extend the theory of DAE to POMDPs, providing a generalized return decomposition.

- We address the problem of increased computational cost of Off-policy DAE by modeling stochastic transitions in a low dimensional embedding space.

- We evaluate our approach using the Arcade Learning Environment (Bellemare et al., 2013), and show that it (1) scales with the capacity of the function approximator, and (2) achieves performance comparable to Rainbow DQN (Hessel et al., 2018) while only using 10% of the data. In addition, we perform extensive ablation studies to understand the contribution of each component.

## 2 BACKGROUND

In the present work, we consider a discounted POMDP defined by the tuple $(\mathcal{S}, \mathcal{A}, T, \Omega, \mathcal{O}, r, \gamma)$ (Kaelbling et al., 1998), where $\mathcal{S}$ is the state space, $\mathcal{A}$ is the action space, $T(s, a, s')$ denotes the transition probability from state $s$ into state $s'$ after taking action $a$, $\Omega$ is the observation space, $\mathcal{O}(s, o)$ denotes the probability of observing $o \in \Omega$ in state $s$, $r(s, a)$ denotes the reward received by the agent after taking action $a$ in state $s$, and $\gamma \in [0, 1)$ denotes the discount factor. When the context is clear, we shall simply denote $T(s, a, s')$ by $p(s'|s, a)$, and $\mathcal{O}(s, o)$ by $p(o|s)$. We consider the case where $\mathcal{S}$, $\mathcal{A}$, and $\Omega$ are finite. An agent in a POMDP cannot directly observe the states, but only the observations emitted from the state through $\mathcal{O}$. We focus on the infinite-horizon discounted setting, where the goal of an agent is to find a policy $\pi$ that maximizes the expected return $J(\pi) = \mathbb{E}_\pi \left[ \sum_{t=0}^{\infty} \gamma^t r(s_t, a_t) \right]$.

In fully observable environments, one can estimate the state(-action) value function $V^\pi(s)$ or $Q^\pi(s, a)$ as the states are observed directly. In POMDPs, however, agents do not observe states directly, and have to estimate the values based on the observed history (information vector) $h_t = (o_0, a_0, r_0, o_1, ..., o_t)$ (Bertsekas, 2012). As such, their counterparts in POMDPs are defined by:

$$V^\pi(h_t) = \mathbb{E}_\pi \left[ \sum_{t'=0}^{\infty} \gamma^{t'} r_{t+t'} \Bigg| h_t \right], \quad Q^\pi(h_t, a_t) = \mathbb{E}_\pi \left[ \sum_{t'=0}^{\infty} \gamma^{t'} r_{t+t'} \Bigg| h_t, a_t \right]. \quad (1)$$

### 2.1 DIRECT ADVANTAGE ESTIMATION

Aside from $Q$ and $V$, another function of interest is the advantage function defined by $A^\pi(s, a) = Q^\pi(s, a) - V^\pi(s)$ (Baird, 1995). Pan et al. (2022) proposed Direct Advantage Estimation (DAE) to estimate the advantage function by minimizing

$$\mathcal{L}(\hat{A}, \hat{V}) = \mathbb{E}_\pi \left[ \left( \sum_{t=0}^{n-1} \gamma^t (r_t - \hat{A}_t) + \gamma^n \hat{V}_{\text{target}}(s_n) - \hat{V}(s_0) \right)^2 \right] \quad \text{s.t.} \ \mathbb{E}_\pi[\hat{A}(s, a)|s] = 0, \quad (2)$$

where $\hat{V}_{\text{target}}$ is a given bootstrapping target, $r_t = r(s_t, a_t)$, and $\hat{A}_t = \hat{A}(s_t, a_t)$. The constraint enforces the centering property of the advantage function (i.e., $\mathbb{E}_\pi[A^\pi(s, a)|s] = 0$). The minimizer of $\mathcal{L}(\hat{A}, \hat{V})$ can be viewed as a multi-step estimate of $(A^\pi, V^\pi)$, as the objective includes multiple steps of unbiased rewards. One limitation of DAE is that it is on-policy, that is, the behavior policy ($\mathbb{E}_\pi$ in the objective) has to be the same as the target policy ($\mathbb{E}_\pi$ in the constraint). Pan & Schölkopf (2024) extended DAE to off-policy settings, by showing that if we view stochastic transitions as actions from an imaginary agent (nature), then the return of a trajectory can be decomposed using the advantage functions from both agents:

$$\sum_{t=0}^{\infty} \gamma^t r(s_t, a_t) = \sum_{t=0}^{\infty} \gamma^t \left( A^\pi(s_t, a_t) + B^\pi(s_t, a_t, s_{t+1}) \right) + V^\pi(s_0), \quad (3)$$

where $B^\pi(s_t, a_t, s_{t+1}) = \gamma V^\pi(s_{t+1}) - \gamma \mathbb{E}_{s' \sim p(\cdot|s_t, a_t)}[V^\pi(s')]$ is the advantage function of nature, which was referred to as *luck* as it quantifies how much of the return is caused by the randomness of the environment. This decomposition generalizes DAE into off-policy settings by incorporating $\hat{B}$ into the objective function (Equation 2):

$$\mathcal{L}(\hat{A}, \hat{B}, \hat{V}) = \mathbb{E}_\mu \left[ \left( \sum_{t=0}^{n-1} \gamma^t (r_t - \hat{A}_t - \hat{B}_t) + \gamma^n \hat{V}(s_n) - \hat{V}(s_0) \right)^2 \right]$$

$$\text{subject to} \ \begin{cases} \mathbb{E}_\pi[\hat{A}(s, a)|s] = 0 \\ \mathbb{E}_{s' \sim p(\cdot|s, a)}[\hat{B}(s, a, s')] = 0 \end{cases}. \quad (4)$$

Contrary to Equation 2, the behavior policy ($\mathbb{E}_\mu$) and the target policy ($\pi$ in the constraint) need not be equal. Intuitively, $\hat{A}$ and $\hat{B}$ can be viewed as corrections for stochasticity originating from the policy and the transitions, respectively. Under mild assumptions on the coverage of $\mu$, one can show that $(A^\pi, B^\pi, V^\pi)$ is the unique minimizer of this objective function, suggesting that we can

perform multi-step off-policy policy evaluation by minimizing the empirical version of this objective function. One benefit of this objective is that it does not require importance sampling to correct for off-policy data, which can lead to unbounded variance. However, this approach has some limitations:

- It only applies to fully observable MDPs.

- Enforcing the $\hat{B}$ constraint in Equation 4 requires estimating the transition probability $p(s'|s, a)$, which can be computationally expensive when the state space is high-dimensional (e.g., images). Pan & Schölkopf (2024) reported approximately 7-fold increase in runtime due to learning the transition probability.

We address these issues in Section 3.

## 3 RETURN DECOMPOSITION IN POMDPS

The key observation of Pan & Schölkopf (2024) is that the return can be decomposed using two different advantage functions (Equation 3). Here, we show that such a decomposition also exists in POMDPs.

Firstly, we can define the advantage function in POMDPs by

$$A^\pi(h_t, a_t) = Q^\pi(h_t, a_t) - V^\pi(h_t) = \mathbb{E}_\pi\left[\sum_{t'=0}^\infty \gamma^{t'} r_{t+t'}\,\middle|\, h_t, a_t\right] - \mathbb{E}_\pi\left[\sum_{t'=0}^\infty \gamma^{t'} r_{t+t'}\,\middle|\, h_t\right]. \quad (5)$$

Similar to its counterpart in MDPs, this function also satisfies the centering property, namely $\sum_{a\in\mathcal{A}} \pi(a|h_t)A^\pi(h_t, a) = 0$. The next question is how we can similarly define $B^\pi$ such that the return can be decomposed, and whether this function also satisfies the centering condition. We proceed by examining the difference between the return and the sum of the advantage function along a given trajectory $(o_0, a_0, r_0, o_1, a_1, r_1, ...)$

$$\sum_{t=0}^\infty \gamma^t r_t - \left(\sum_{t=0}^\infty \gamma^t A^\pi(h_t, a_t) + V^\pi(h_0)\right) = \sum_{t=0}^\infty \gamma^t \left(r_t + \gamma V^\pi(h_{t+1}) - Q^\pi(h_t, a_t)\right). \quad (6)$$

This equation suggests that we can define $B^\pi$ for POMDPs by:

$$B^\pi(h_t, a_t, h_{t+1}) = r_t + \gamma V^\pi(h_{t+1}) - Q^\pi(h_t, a_t) \quad (7)$$

Recall that $h_{t+1}$ is simply the concatenation of $h_t$ and $(a_t, r_t, o_{t+1})$, meaning that we can rewrite $B^\pi(h_t, a_t, h_{t+1})$ as $B^\pi(h_t, a_t, r_t, o_{t+1})$. We thus see that this $B^\pi$ also satisfies a (slightly different) centering property, namely,

$$\mathbb{E}_{(r_t, o_{t+1})\sim p(\cdot|h_t, a_t)}\left[B^\pi(h_t, a_t, r_t, o_{t+1})\right] = 0. \quad (8)$$

Essentially, this equation differs from its MDP counterpart by the variables that are being marginalized. In POMDPs, since the agent cannot observe the underlying state, we have to marginalize over the observed variables after taking an action (i.e., the immediate reward and the next observation).

Finally, we arrive at the following generalization:

**Proposition 1.** *Given behavior policy $\mu$, target policy $\pi$, and backup length $n > 0$. $(A^\pi, B^\pi, V^\pi)$ is a minimizer of*

$$\mathcal{L}(\hat{A}, \hat{B}, \hat{V}) = \mathbb{E}_\mu\left[\left(\sum_{t'=0}^{n-1} \gamma^{t'}\left(r_{t+t'} - \hat{A}_{t+t'} - \hat{B}_{t+t'}\right) + \gamma^n \hat{V}(h_{n+t}) - \hat{V}(h_t)\right)^2\right] \quad (9)$$

$$\text{subject to } \begin{cases} \mathbb{E}_{a\sim\pi(\cdot|h)}[\hat{A}(h, a)] = 0 & \forall h \in \mathcal{H} \\ \mathbb{E}_{(r,o')\sim p(\cdot|h,a)}[\hat{B}(h, a, r, o')] = 0 & \forall (h, a) \in \mathcal{H} \times \mathcal{A} \end{cases},$$

*where $\mathcal{H}$ is the set of all trajectories of the form $(o_0, a_0, r_0, ...o_t)$, $\hat{A}_t = \hat{A}(h_t, a_t)$, and $\hat{B}_t = \hat{B}(h_t, a_t, r_t, o_{t+1})$. Furthermore, if $p_\pi(h) > 0 \implies p_\mu(h) > 0$ holds for all $h$ (i.e., the behavior policy has a larger coverage), then the minimizer is unique at trajectories covered by $\pi$.*

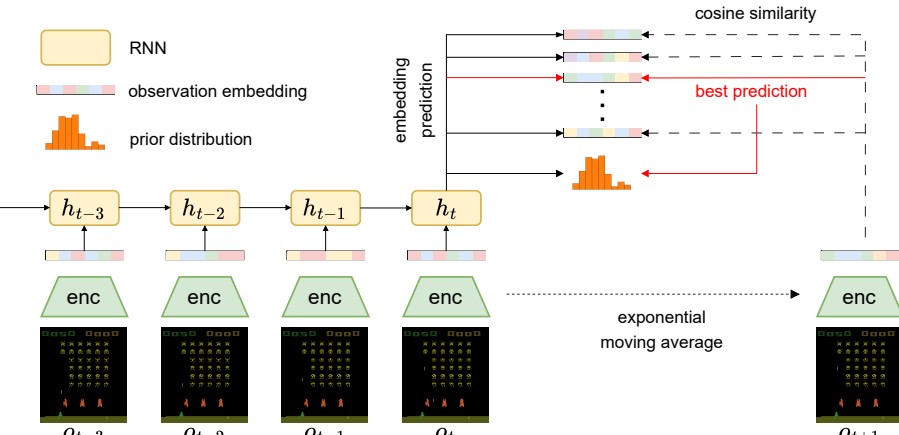

Figure 1: The latent dynamics model first embeds observations ($o_t$) into low dimensional embeddings ($x_t$), which are then processed by an RNN to capture the information vectors ($h_t$) (for illustrative purpose, we omit conditioning on previous actions and rewards). At each time-step, the model makes $|\mathcal{Z}|$ predictions ($\hat{x}$) of the next embedding along with the prior distribution ($p_\phi(\cdot|h,a)$) to capture the stochasticity. During training, the embedding prediction is compared to the embedding of the next observation generated by the exponential moving average encoder, and gradients only propagate through the best prediction with the corresponding index used to train the categorical prior distribution.

See Appendix A for a proof. Note that, slightly different from the MDP version, we do not require additional coverage regarding actions because $h$ already encodes past actions. At its core, Proposition 1 differs from its MDP counterpart (Equation 4) by simply replacing states with histories, and transition probabilities with conditional densities of the observed variables (in the $\hat{B}$ constraint). This is a consequence of the fact that POMDPs can be reformulated as MDPs using information vectors (Bertsekas, 2012). Like DAE, this can be seen as an off-policy multi-step method for value approximation, as the objective function includes $n$ steps of unbiased rewards. Deploying this method in practice, however, is non-trivial due to the constraints. The $\hat{A}$ constraint can be easily enforced upon a given function approximator $f(h,a)$ for a given policy $\pi$ by constructing $\hat{A}(h,a) = f(h,a) - \sum_{a \in \mathcal{A}} f(h,a)\pi(a|h)$ (Wang et al., 2016). Enforcing the $\hat{B}$ constraint is more challenging due to its dependency on $p(r,o'|h,a)$, which is typically unknown to the agent. We discuss how to efficiently approximate this constraint using latent dynamics models below.

## 3.1 DISCRETE LATENT DYNAMICS MODEL FOR CONSTRAINT APPROXIMATION

In the original Off-policy DAE implementation (Pan & Schölkopf, 2024), the $\hat{B}$ constraint was approximated by first encoding transitions $(h,a,r,o')$[1] into a small discrete latent space $z \in \mathcal{Z}$ using a conditional variational autoencoder (CVAE) (Kingma & Welling, 2013; Sohn et al., 2015), and constructing $\hat{B}(h,a,r,o')$ from a given function approximator $g(h,a,z)$ by:

$$\hat{B}(h,a,r,o') = \mathbb{E}_{z \sim q_\phi(\cdot|h,a,r,o')}[g(h,a,z)] - \mathbb{E}_{z \sim p_\phi(\cdot|h,a)}[g(h,a,z)], \tag{10}$$

where $q_\phi(\cdot|h,a,r,o')$ is the approximated posterior (encoder), $p_\phi(\cdot|h,a)$ is the prior, and $\phi$ is the parameters of the CVAE. By using discrete latent variables, the expectations with respect to $z$ can be computed efficiently in practice. It then follows that $\mathbb{E}_{(r,o') \sim p(\cdot|h,a)}[\hat{B}(h,a,r,o')] \approx 0$. This approach, however, can be computationally expensive if observations are high dimensional due to the need to reconstruct observations.

---

[1]We adopt the POMDP setting here for consistency, but note that this was originally developed for MDPs.

To reduce computational overhead, we propose to learn a discrete dynamics model purely in the embedding space[2] (see Figure 1). This is achieved by first embedding observations into a low dimensional vector $x = \texttt{enc}(o) \in \mathbb{R}^d$ (with $d \ll \dim(\Omega)$), where $\texttt{enc}$ denotes the encoder (e.g., a convolutional network), and learning to predict $x_{t+1} = \texttt{enc}(o_{t+1})$ from the observed history $(h_t, a_t)$. This approach is similar to the self-predictive representation (SPR) (Schwarzer et al., 2020); however, SPR only produces a single prediction, which cannot capture stochastic transitions. We address this by combining SPR with the Winner-Takes-All (WTA) loss (Lee et al., 2015; Guzman-Rivera et al., 2012), which was shown to be useful for modeling stochastic predictions. More specifically, we combine them by: (1) making $|\mathcal{Z}|$ predictions of the next embedding (note that $|\mathcal{Z}|$ is an integer since we are using discrete latent variables), and (2) minimizing only the best prediction. This results in the following objective function:

$$\mathcal{L}_{\text{rec}} = \sum_{z \in \mathcal{Z}} \mathbb{I}\big[z = \arg\min_i ||\hat{x}_{t+1,i}(h_t, a_t) - x_{t+1}||\big] ||\hat{x}_{t+1,z}(h_t, a_t) - \texttt{sg}(x_{t+1})||^2, \qquad (11)$$

where $\mathbb{I}$ is the indicator function, and $\texttt{sg}$ denotes stop-gradient. Intuitively, this can be seen as performing $k$-means clustering (with $k = |\mathcal{Z}|$) in the embedding space with centroids $\hat{x}_{\cdot,z}$ (Rupprecht et al., 2017). The WTA loss is known to be difficult to train as the gradient only propagates through the best prediction, which can sometimes lead to collapse of predictions. As such, in practice, we use an annealing procedure similar to the evolving WTA (Makansi et al., 2019), where the indicator function is replaced by a soft weighting (see Appendix B for details). Next, note that the objective is equivalent to a conditional vector-quantized VAE (VQ-VAE) (Van Den Oord et al., 2017), with posterior $q_\phi(z|h_t, a_t, x_{t+1}) = \mathbb{I}[z = \arg\min_i ||\hat{x}_{t+1,i}(h_t, a_t) - x_{t+1}||]$, and codebook $\hat{x}_{t+1,z}(h_t, a_t)$ that are dependent on the information vector $h_t$. Consequently, we can learn the prior by minimizing the KL-divergence between the prior $p_\phi(z|h_t, a_t)$ and the posterior $q_\phi(z|h_t, a_t, x_{t+1})$. With this CVAE, we can then approximate the $\hat{B}$ constraint using Equation 10. Note that the $\hat{B}$ constraint indicates that we should also consider stochasticity from the rewards. This can be achieved by adding another reward reconstruction term into Equation 11.

In practice, we find that using shallow multilayer perceptrons (MLPs) to model the dynamics already achieves strong empirical performance with negligible computational overhead compared to other parts of the system. In addition, we found it possible to learn the RL objective (Equation 9) and the dynamics model jointly end-to-end to further reduce computational overhead compared to learning them separately as done by Pan & Schölkopf (2024).

## 4 Experiments

We examine the performance of the POMDP version of DAE using 47 environments[3] from the Arcade Learning Environment (ALE) (Bellemare et al., 2013), which includes environments with diverse dynamics and various degrees of partial observability.

We use the same environment setting as the Dopamine baselines (Castro et al., 2018), which largely follows the modern evaluation protocols proposed by Machado et al. (2018), including the use of sticky actions (repeat previous action with a certain probability) and discarding end-of-life signals. Note that while sticky actions were originally proposed to inject stochasticity into the environments, they also introduce additional partial observability due to their dependencies on previous actions.

We evaluate our method using a DQN-like (Mnih et al., 2015) agent with some modifications, which we briefly summarize: (1) **Recurrent Architecture**: We use an LSTM (Hochreiter & Schmidhuber, 1997) after the convolutional encoder to process sequences of observations. Aside from observations, we also feed previous actions and rewards into the LSTM to model the full history. Similar to R2D2 (Kapturowski et al., 2018), we also store the recurrent states in the replay buffer and include a short burn-in sequence to initialize the LSTM states. (2) **DAE objective**: We replace the 1-step Q-learning objective with the multi-step DAE objective (Equation 9, we set $n=16$ by default), and use three separate MLPs on top of the LSTM to model $\hat{A}$, $\hat{B}$, and $\hat{V}$. (3) **Discrete Latent Dynamics Model**: We use three additional MLPs on top of the LSTM to estimate the next observation

---

[2]We will refer to the space of encoded observations as the embedding space, and $\mathcal{Z}$ as the latent space of the CVAE to avoid confusion.

[3]We exclude hard exploration games such as Montezuma's Revenge, as they typically require specialized exploration strategies, and often have low predictive power on the overall performance (Aitchison et al., 2023).

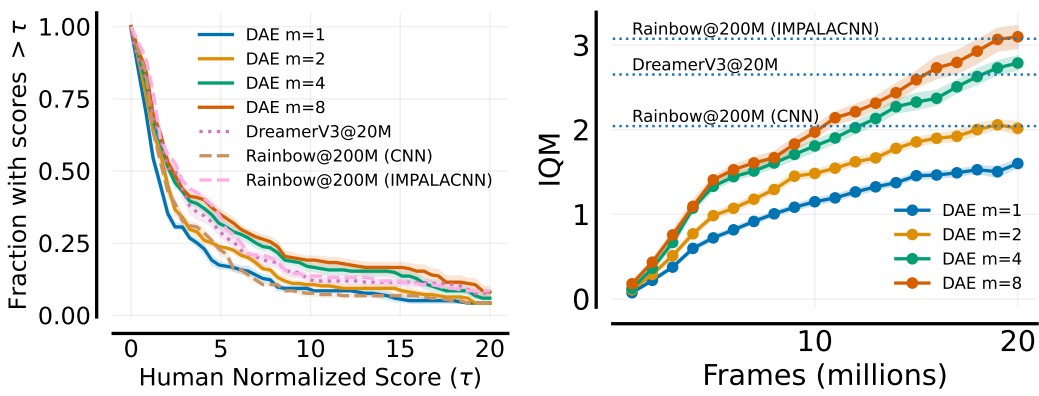

Figure 2: Performance profile (left) and sample efficiency (right) of DAE.

embedding $\hat{x}_{t+1,z}(h_t, a_t)$, the immediate reward $p(r_t|h_t, a_t)$, and the prior distribution $p(z|h_t, a_t)$ for approximating the $\hat{B}$ constraint. Similar to SPR, we use an exponential moving average of the online network as the target network to generate the next observation embeddings for the dynamics model training. This target network is also used to construct smoothly changing target policies (softmax of the advantage) and value bootstrapping targets for the DAE objective, which was found to be important for DAE (Pan & Schölkopf, 2024). (4) **Deeper Network**: We replace the shallow three-layer convolutional network used in the original DQN by the 15-layer deep residual network proposed by Espeholt et al. (2018) (denoted IMPALACNN below), which was found to enjoy better scalability and improved sample efficiency (Schwarzer et al., 2023). The CNN-LSTM backbone is shared for both the value heads and the dynamics heads to reduce computational overhead. For more details, we refer the reader to Appendix B. In terms of RL, our agent can be viewed as a DQN variant with (a) **POMDP correction** and (b) **multi-step off-policy learning** (enabled by DAE). Below, we show how these changes affect the performance of the agent.

In the following experiments, we train our method for 20 million frames (5 million environment steps due to frame-skipping), and evaluate the agent every 1 million frames by averaging the cumulative scores of 50 episodes. We follow the protocol of Agarwal et al. (2021) and report the interquartile means (IQM) and performance profiles, along with 95% bootstrap confidence intervals aggregated over 5 random seeds and 47 environments.

**Scalability and Sample Efficiency** It was previously shown that naively scaling up the capacity of the function approximator does not always translate to an increase in performance (Obando-Ceron et al., 2024a). On the other hand, DAE was shown to be easily scalable in the on-policy setting (Pan et al., 2022). Here, we examine the scalability of DAE in the off-policy setting by increasing the width (multiplied by $m$) of the IMPALACNN backbone. We compare DAE to three baselines: (1) Dopamine Rainbow DQN (Castro et al., 2018; Hessel et al., 2018)[4] (2) A scaled-up version of Rainbow DQN using IMPALACNN, and (3) DreamerV3 (Hafner et al., 2023). Both (1) and (2) represent classical frame-stacking model-free baselines, whereas (3) is a more recent recurrent model-based approach closer to our agent. For baselines (1) and (2), we use the scores reported by Castro et al. (2018), which were trained for 200 million frames. For (3), we retrain DreamerV3 using the preset 50M parameter network (the default network has 400M parameters), which has a similar number of parameters to our $m=8$ variant, to establish a closer comparison (see Appendix B.5 for more details). Figure 2 summarizes the results. We see that the $m=2$ variant already performs similarly to Rainbow while using only 10% of the training frames, and by scaling up the network to $m=8$, we achieve performance comparable to Rainbow with IMPALACNN. Similarly, we find that DAE is competitive with DreamerV3 at $m=4$, and is even more sample efficient at $m=8$.

Next, we perform ablation studies to better understand the contribution of the modifications. To limit computational cost, we use only the $m = 4$ model. Figure 3 and Table 1 summarize the results.

---

[4]Dopamine Rainbow DQN is a modern reimplementation of the Rainbow DQN, which includes 3 core improvements ($n$-step backup, prioritized replay, and distributional RL) from the original implementation.

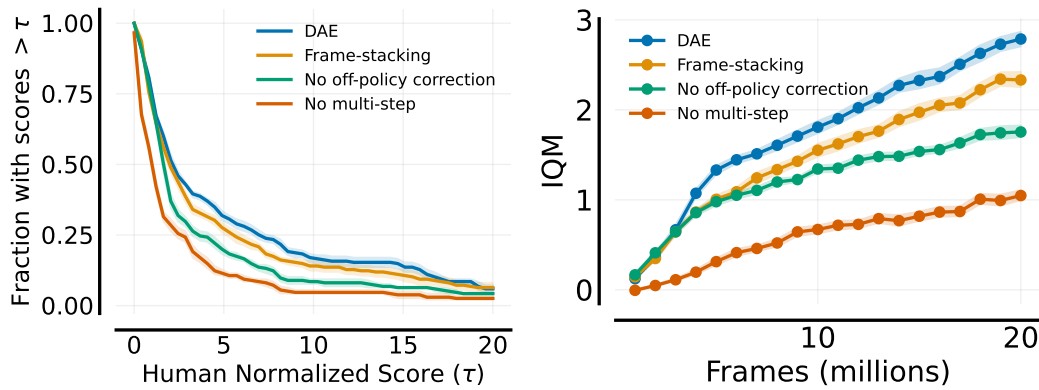

Figure 3: Performance profile (left) and sample efficiency (right) of the ablation study. For each ablation, we remove the corresponding component from the base DAE agent.

**Multi-step Learning** Multi-step learning speeds up learning, reduces bias from bootstrapping, and was found to stabilize training (Hernandez-Garcia & Sutton, 2019; Van Hasselt et al., 2018). However, it also increases the variance of value updates, and choosing the backup length $n$ can be seen as a bias-variance tradeoff (Kearns & Singh, 2000). Here, we compare multi-step learning ($n = 16$) to single-step learning ($n = 1$). From the learning efficiency curve (Figure 3, right), we see that multi-step learning sig-

Table 1: Effect of each component.

| Ablation | IQM |
|---|---|
| DAE | 2.79 |
| Frame-Stacking | 2.33 (-0.46) |
| No off-policy correction | 1.75 (-1.04) |
| No multi-step | 1.05 (-1.74) |

nificantly improves sample efficiency, and is, in fact, the most important component in this ablation study, accounting for a decrease of 1.74 in the IQM score. This suggests that the bias introduced by bootstrapping significantly exceeds the variance from the multi-step learning in this case.

**Off-policy Correction** In previous studies, multi-step learning was often used without proper off-policy corrections (Van Hasselt et al., 2018; Hessel et al., 2018; Kapturowski et al., 2018; Hernandez-Garcia & Sutton, 2019). Here, we demonstrate the importance of off-policy correction. Similar to Pan & Schölkopf (2024), we partially disable off-policy corrections by setting $\hat{B} \equiv 0$ during training, which can be seen as ignoring stochasticity from the environment (note that $B^\pi \equiv 0$ for deterministic environments).[5] Consistent with prior work, we find that, even without off-policy corrections, multi-step learning still significantly outperforms single-step learning. However, off-policy correction further improves the performance of our agent. This also indicates that the learned latent dynamics model can well approximate the dynamics, since the $\hat{B}$ constraint hinges on this approximation.

**Frame-Stacking** Frame-stacking has been the standard approach to approximate the ALE environments as MDPs since its introduction by Mnih et al. (2015). However, previous works have demonstrated that frame-stacking is not enough to fully capture the partial observability of the ALE (Kapturowski et al., 2018). Here, we demonstrate the effectiveness of the POMDP correction by replacing the recurrent layer by a single-layer MLP with a similar number of parameters. Consistent with prior work (Kapturowski et al., 2018; Hausknecht & Stone, 2015), we find that frame-stacking is suboptimal and accounts for $\sim 16\%$ of the performance degradation (-0.46 in IQM score), indicating the importance of the POMDP correction. Aside from performance, we also see qualitative differences in the entropy of the prior distribution of the dynamics model. Note that since the ALE is deterministic by design, uncertainties of the next observation prediction comes primarily from partial observability of the environment (another source is sticky-actions, which also introduces stochasticity). Figure 4 compares the entropy of the prior distributions of the dynamics models and the learning curves between BattleZone and Pong. In BattleZone, an agent has to control a tank in a 3D environment with limited views of its surroundings in third-person. In contrast, Pong

[5]The typical multi-step method is more aggressive and equivalent to enforcing both $\hat{A} \equiv 0$ and $\hat{B} \equiv 0$.

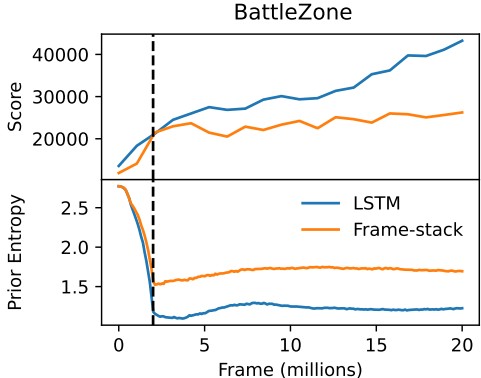
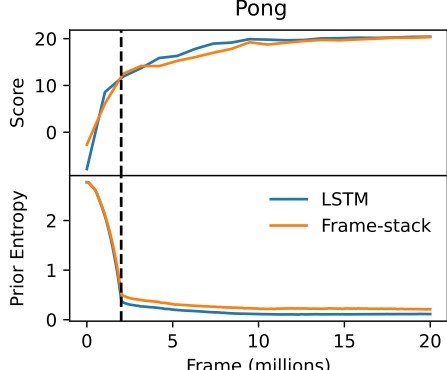

Figure 4: Entropy of the prior distribution $p(z|h, a)$ during training in BattleZone (left, more partially observable) and Pong (right, nearly fully observable). Dashed lines indicate the end of WTA annealing.

is almost fully observable except for the velocities of the objects (ball and paddles), which can be inferred from the past few frames. Here, we see that the LSTM agent converges to a much lower entropy compared to the frame-stacking agent in BattleZone, suggesting that the next observations have dependencies beyond the most recent 4 frames that cannot be utilized by the frame-stacking agent. On the other hand, both the LSTM and the frame-stacking agents performed similarly in Pong with very low entropy, indicating that the environment has a low degree of partial observability. In Appendix B.6, we further investigate the link between changes in the entropy and changes in the performance, and provide additional evidence that partial observability degrades the performance of frame-stacking agents.

**Discrete Latent Dynamics Model** Our latent dynamics model leverages multiple predictions to model the stochasticity of the environments. To assess the robustness of this approach, we vary the number of predictions $|\mathcal{Z}|$ and report performance profile in Figure 5. We find that $|\mathcal{Z}| = 8$ and $|\mathcal{Z}| = 16$ yield nearly identical results, with noticeable degradation only when $|\mathcal{Z}|$ is reduced to 4. This indicates that, while the stochasticity cannot be ignored, in most environments it can be captured with relatively few modes, and our approach remains robust to the choice of $|\mathcal{Z}|$ provided it is not too small.

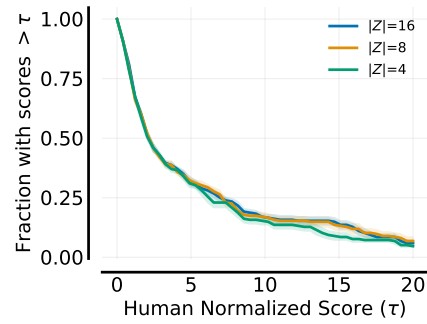

Figure 5: Effect of $|\mathcal{Z}|$.

To summarize the ablation studies, we observe from the performance profile (Figure 3, left) that the curves are almost dominated by the base agent, suggesting that the corrections are not environment-specific, but rather general algorithmic improvements. Additional per-environment results can be found in Appendix B.6.

## 5 RELATED WORK

**Advantage Estimation** Baird (1994) first introduced the advantage function and the algorithm *advantage updating* to solve continuous time RL problems. Later, Kakade & Langford (2002) showed that the performance difference between two policies can be described using the advantage function, which became the foundation of various modern policy optimization algorithms. More recently, Schulman et al. (2015) proposed Generalized Advantage Estimation (GAE), which utilizes TD($\lambda$) (Sutton, 1988) to perform on-policy multi-step estimates of the advantage function, and demonstrated its effectiveness in continuous control settings. Wang et al. (2016) proposed dueling network, an extension of DQN, which parametrized $Q_\theta = V_\theta + A_\theta$ and showed that this parametriza-

tion improves the performance of the original DQN. Tang et al. (2023) proposed VA learning, which uses a similar decomposition, but updates $V$ and $A$ separately, and showed its convergence properties in the tabular setting and effectiveness when applied to deep RL settings. Pan et al. (2022) proposed DAE to perform on-policy multi-step estimation of the advantage function, and was later shown to be equivalent to learning control variates for policy evaluation (Pan & Schölkopf, 2025). DAE was later generalized to the off-policy setting by Pan & Schölkopf (2024), and the present work extends its domain to POMDPs and improves its computational efficiency.

**Partial Observability** POMDPs provide a general framework for studying decision making with incomplete states (Åström, 1965). In RL, POMDPs are usually solved by first converting them into MDPs using belief states or information vectors (Bertsekas, 2012; Kaelbling et al., 1998). In deep RL, common approaches include frame-stacking (Mnih et al., 2015), or modeling the histories directly (Kapturowski et al., 2018; Hausknecht & Stone, 2015; Hafner et al., 2023).

**Latent Dynamics Model** Learning dynamics models in the latent space is a promising approach to model-based RL (Ha & Schmidhuber, 2018; Han et al., 2019; Schrittwieser et al., 2020; Hafner et al., 2023; Antonoglou et al., 2021). Similar ideas have also been explored using bisimulation metrics (Ferns et al., 2004; Zhang et al., 2020), and were shown to be effective in learning representations that are effective for downstream tasks. However, learning dynamics models in the latent space is prone to collapse, and it is common to rely on either reconstructing the observations or self-supervision to learn meaningful representations (Anand et al., 2021; Deng et al., 2022). In the present work, we combined self-supervised learning methods (Schwarzer et al., 2020; Grill et al., 2020) and the winner-takes-all loss (Makansi et al., 2019; Rupprecht et al., 2017) to overcome the collapsing problem and reduce computational overhead of learning high dimensional models.

**Scaling Deep RL** Scaling has been central to progress in deep learning, where larger models were shown to yield better performance (Hestness et al., 2017; Henighan et al., 2020; Zhai et al., 2022). However, scaling in deep reinforcement learning (RL) is more challenging due to unstable training dynamics, and often requires additional regularization or architectural changes (Obando-Ceron et al., 2024b; Schwarzer et al., 2023; Nauman et al., 2024; Castanyer et al., 2025).

# 6 DISCUSSION

In the present work, we extended DAE for POMDPs and addressed its high computational complexity by using discrete latent dynamics models. This opens up possibilities of using DAE to build sample-efficient RL agents for challenging real-world domains, where partial observability and high-dimensional observations are common. Through experiments in the ALE with a modified DQN agent, we demonstrated that DAE is sample-efficient and scalable, and verified the effectiveness of the proposed corrections.

We emphasize that, although our method learns transition models for DAE, they are not used for rollouts as in typical model-based deep RL. Instead, they serve only to perform off-policy corrections (enforcing the $\hat{B}$ constraint in the DAE objective) during value learning. From this perspective, our approach is more model-free than model-based. Moreover, its ability to scale easily without additional tuning suggests that one of the challenges in scaling up value-based model-free methods like DQN may lie in the objective function itself, which might not be well suited to current deep learning architectures. In contrast, the DAE objective appears inherently easier to optimize. A more detailed investigation of this hypothesis is left for future work.

Finally, we note some limitations: (1) DAE requires learning the transition probabilities to approximate constraints for the objective. While we demonstrated the effectiveness of learning latent dynamics models to achieve this, doing so inevitably introduces additional hyperparameters (e.g., network architectures of the dynamics models) and complicates the implementation. Interestingly, this type of problem falls under a more general setting known as conditional moment restriction, and is an active research area (Newey, 1990; Bennett et al., 2019; Muandet et al., 2020; Kremer & Schölkopf, 2024). One direction for future work is to explore more robust and efficient alternatives to approximate the constraints. (2) We explored scaling of the convolutional layers of the image encoders, but it remains an open question how scaling other components, such other parts of the function approximator or number of updates, might impact performance. A more systematic study of scaling across the full architecture could yield further insights.

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

## A   PROOF OF PROPOSITION 1

**Proposition 1.** *Given behavior policy $\mu$, target policy $\pi$, and backup length $n > 0$. $(A^\pi, B^\pi, V^\pi)$ is a minimizer of*

$$
\mathcal{L}(\hat{A}, \hat{B}, \hat{V}) = \mathbb{E}_\mu \left[ \left( \sum_{t'=0}^{n-1} \gamma^{t'} \left( r_{t+t'} - \hat{A}_{t+t'} - \hat{B}_{t+t'} \right) + \gamma^n \hat{V}(h_{n+t}) - \hat{V}(h_t) \right)^2 \right]
\tag{9}
$$

$$
\text{subject to} \begin{cases} \mathbb{E}_{a \sim \pi(\cdot|h)}[\hat{A}(h,a)] = 0 & \forall h \in \mathcal{H} \\ \mathbb{E}_{(r,o') \sim p(\cdot|h,a)}[\hat{B}(h,a,r,o')] = 0 & \forall (h,a) \in \mathcal{H} \times \mathcal{A} \end{cases},
$$

*where $\mathcal{H}$ is the set of all trajectories of the form $(o_0, a_0, r_0, ...o_t)$, $\hat{A}_t = \hat{A}(h_t, a_t)$, and $\hat{B}_t = \hat{B}(h_t, a_t, r_t, o_{t+1})$. Furthermore, if $p_\pi(h) > 0 \implies p_\mu(h) > 0$ holds for all $h$ (i.e., the behavior policy has a larger coverage), then the minimizer is unique at trajectories covered by $\pi$.*

*Proof.* First, consider the MDP induced by the POMDP (with $\mathcal{S} = \mathcal{H}$) (Bertsekas, 2012). The theorem is then a direct result of applying Off-policy DAE (Pan & Schölkopf, 2024) to this MDP. □

Remark: The original proof of Off-policy DAE assumes that the reward function is deterministic, which can be violated when converting POMDPs into MDPs. As such, our definition of $B^\pi(s,a,r,s') = r + \gamma V^\pi(s') - \mathbb{E}_{(r',s'') \sim p(\cdot|s,a)}[r' + \gamma V^\pi(s'')]$ (in a fully observable MDP) differs slightly from the original one $B^\pi(s,a,s') = \gamma V^\pi(s') - \mathbb{E}_{s'' \sim p(\cdot|s,a)}[\gamma V^\pi(s'')]$.

## B   EXPERIMENT DETAILS & ADDITIONAL RESULTS

### B.1   PSEUDOCODE AND ADDITIONAL IMPLEMENTATION DETAILS

We provide the pseudocode in Algorithm 1. For illustrative purpose, the pseudocode assumes a single actor during sampling and batch size 1 during training. Below, we discuss some implementation details.

**WTA Training** To avoid the WTA predictions from collapsing, we use a soft loss for the reconstruction by including $\epsilon_{\text{WTA}} \geq 0$ into the posterior construction. In practice, $\epsilon_{\text{WTA}}$ is linearly annealed from 1 to 0 in the early stage of training. More specifically, the posterior becomes

$$p(z|h_t, a_t, x_{t+1}) = \begin{cases} 1 - \epsilon_{\text{WTA}} + \frac{\epsilon_{\text{WTA}}}{|\mathcal{Z}|} & \text{if } z = \arg\min_z \|\hat{x}_{t+1,z} - x_{t+1}\| \\ \frac{\epsilon_{\text{WTA}}}{|\mathcal{Z}|} & \text{otherwise} \end{cases},$$

This is similar to the approach proposed by Makansi et al. (2019), which was found to make training less dependent on initialization, except that top-$k$ nearest neighbors were used to construct the posterior. We found the posterior can change rapidly at the beginning of training and lead to instability of $\hat{B}$. As such, we do not include the $\hat{B}$ correction (simply force $\hat{B} \equiv 0$) for the first few steps, and only include it after $\epsilon_{\text{WTA}} \leq 0.75$ (approximately 125000 environment steps). In addition, we multiply the DAE objective by $(1 - \epsilon_{\text{WTA}})$, to prioritize model learning during the early phase of training.

Incorporating stochastic rewards in the transition model was achieved by adding a reward prediction head. In the case of the ALE, we exploit the discreteness of the rewards ($\mathcal{R} = \{-1, 0, 1\}$ due to clipping) and construct the latent space by $\mathcal{Z} = \mathcal{Z}_O \times \mathcal{R}$. This then allows us to decompose the prior and the posterior by $p(z|h, a) = p(z_o|h, a)p(r|h, a)$ and $q(z|h, a, r, o') = q(z_o|h, a, r, o')q(\hat{r}|h, a, r, o')$, respectively. In this case, the posterior $q(\hat{r}|h, a, r, o') = \mathbb{I}(\hat{r} = r)$ is simply the indicator function.

**Target Policy** As pointed out by Pan et al. (2022), having a smoothly changing target policy is crucial to optimizing the DAE objective function. Consequently, we construct the target policy using the softmax of $\hat{A}_{\theta_{\text{EMA}}}$. However, as reward densities can vary drastically between environments and lead to different scales of the advantage function, We additionally learn a temperature parameter $T$ by minimizing $\log T + \beta_{\text{KL}}\text{KL}(\pi_{\text{EMA}}||\pi)$, where both policies $\pi$ and $\pi_{\text{EMA}}$ are softmax policies constructed using the advantage functions (i.e. $\pi = \text{softmax}(\frac{\hat{A}}{T})$). This KL divergence ensures that the online policy $\pi$ does not deviate too much from the target policy $\pi_{\text{EMA}}$, and alleviates the need to tune the temperature manually for each environment. We note that this policy is only used for the DAE objective ($\hat{A}$ constraint), and not for data collection.

Finally, to balance the scales between various objective functions, we multiply the DAE loss by $\beta_V = \frac{1}{\text{Var}(G)}$ ($\text{Var}(G)$ is the variance of the returns, estimated from all trajectories in the replay buffer).

## B.2 ENVIRONMENT SETTING

The environment settings follow the ones used by the Dopamine baseline (Castro et al., 2018) (see Table 2). We use EnvPool (Weng et al., 2022) for efficient implementation of parallelized environments.

Table 2: ALE preprocessing parameters. Blue: Best practice suggested by Machado et al. (2018).

| Parameter | Value |
|---|---|
| Grey-scaling | True |
| Observation Resolution | $84 \times 84$ |
| Frame Stack | 4 |
| Action Repetitions | 4 |
| Reward Clipping | [-1, 1] |
| Terminal on life-loss | False |
| Sticky Action Prob. | 0.25 |
| $\gamma$ (discount factor) | 0.99 |

## B.3 HYPERPARAMETERS

Table 3 summarizes the default hyperparameters used in the experiments. For the learning rate, we found linear warmup to be important, which is likely due to the use of LSTMs that can be unstable in the early stage of training. The batch size indicates the number of trajectories instead of frames, and the number of frames per batch is (backup length + burn-in) $\times$ batch size.

**Algorithm 1** DAE (POMDP)

**Require:** $n$ (backup length), $k$ (burn-in length), $\tau$ (EMA coefficient), `wta_scheduler`, `optimizer`, $\beta_{\text{prior}}, \beta_{\text{rec}}, \beta_{\text{KL}}$

1: Initialize network $f_\theta$
2: $\theta_{\text{EMA}} \leftarrow \theta$
3: $T \leftarrow 1, T_{\text{EMA}} \leftarrow 1$
4: $D = \{\}$                       (replay buffer)
5: $o_0 \leftarrow$ `env.reset()`
6: $h_0 \leftarrow (o_0)$
7: **for** $t = 0, 1, 2, \ldots$ **do**
8:    $\hat{A}_t, H_t \leftarrow f_\theta(h_t)$                        ($\hat{A}$: advantage, $H_t$: RNN state)
9:    $a \leftarrow \epsilon-$`greedy`$(\hat{A}_t)$
10:   $(r, o') \leftarrow$ `env.step`$(a)$
11:   $h_{t+1} \leftarrow (h_t, a, r, o')$
12:   $D \leftarrow D \cup \{(o, a, r, o', H_t)\}$
13:   **if** $t + 1$ mod `steps_per_update` $= 0$ **then**
14:      $\epsilon_{\text{WTA}} \leftarrow$ `wta_scheduler`$(t)$
15:      `traj` $\leftarrow$ sample $(o_i, a_i, r_i, ..., o_{i+n+k})$ and $H_i$ from $D$
16:      $\hat{V}, \hat{A}, \hat{B}, \hat{x}, \hat{p}(\cdot|h, a) \leftarrow f_\theta(\text{traj})$     (computed along the trajectory)
17:      $\hat{V}_{\text{EMA}}, \hat{A}_{\text{EMA}}, x_{\text{EMA}} \leftarrow f_{\theta_{\text{EMA}}}(\text{traj})$
18:      $q_i(z) \leftarrow \begin{cases} 1 - \epsilon_{\text{WTA}} + \frac{\epsilon_{\text{WTA}}}{|\mathcal{Z}|} & z = \arg\min_z \|\hat{x}_{i+1,z} - x_{\text{EMA},i+1}\| \\ \frac{\epsilon_{\text{WTA}}}{|\mathcal{Z}|} & \text{otherwise} \end{cases}$     (posterior)
19:      $\mathcal{L}_{\text{rec}} \leftarrow \sum_{i \geq k} \sum_z q_i(z) \|\hat{x}_{i+1,z} - x_{\text{EMA},i+1}\|^2$     (reconstruction loss)
20:      $\mathcal{L}_{\text{prior}} \leftarrow -\sum_{i>k} \sum_z q_i(z) \log \hat{p}(z|h_i, a_i)$     (prior loss)
21:      **if** $\epsilon_{\text{WTA}} < \epsilon_{\text{cutoff}}$ **then**
22:        $\hat{B}_i \leftarrow \sum_z (q_i(z) - \text{sg}(\hat{p}(z|h_i, a_i))) \hat{B}(h_i, a_i, z)$     ($\hat{B}$ constraint)
23:      **else**
24:        $\hat{B}_i \leftarrow 0$
25:      **end if**
26:      $\pi_{\text{target}} \leftarrow$ `softmax`$(\frac{\hat{A}_{\text{EMA}}}{T_{\text{EMA}}}), \pi \leftarrow$ `softmax`$(\frac{\text{sg}(\hat{A})}{T})$
27:      $\hat{A}_i \leftarrow \hat{A}(h_i, a_i) - \sum_a \hat{A}(h_i, a) \pi_{\text{target}}(h_i, a)$     ($\hat{A}$ constraint)
28:      $\mathcal{L}_{\text{DAE}} \leftarrow \left(\sum_{j=k}^n \gamma^{j-k}(r_{i+j} - \hat{A}_{i+j} - \hat{B}_{i+j}) + \gamma^{n-k+1}\hat{V}_{\text{EMA},i+n+k} - \hat{V}_i\right)^2$
29:      $\mathcal{L}_T \leftarrow \log T + \beta_{\text{KL}}\text{KL}(\pi_{\text{target}}||\pi)$
30:      $\beta_V \leftarrow \frac{1 - \epsilon_{\text{WTA}}}{\text{Var}_D(G)}$
31:      $\theta, T \leftarrow$ `optimizer`$(\beta_V \mathcal{L}_{\text{DAE}} + \beta_{\text{prior}}\mathcal{L}_{\text{prior}} + \beta_{\text{rec}}\mathcal{L}_{\text{rec}} + \mathcal{L}_T)$
32:      $\theta_{\text{EMA}} \leftarrow \tau\theta_{\text{EMA}} + (1 - \tau)\theta$
33:      $T_{\text{EMA}} \leftarrow \tau T_{\text{EMA}} + (1 - \tau)T$
34:   **end if**
35: **end for**

Table 3: Default hyperparameters for the experiments.

| Parameter | Value |
|---|---|
| Replay buffer size | 1000000 |
| Minimum Steps before training | 20000 |
| Number of parallel actors | 16 |
| $\epsilon$ (training) | Linearly annealed from 1 to 0.01 in the first 1M steps |
| $\epsilon$ (evaluation) | 0.001 |
| Optimizer | Adam (Kingma & Ba, 2014) |
| Learning rate | Linear warmup from 0 to $1.25 \times 10^{-4}$ in the first 100000 steps then linearly annealed to $1.25 \times 10^{-5}$ throughout training |
| Adam $\beta$ | (0.9, 0.95) |
| Adam $\epsilon$ | $10^{-6}$ |
| Replay ratio ($\frac{\text{Gradient updates}}{\text{Environment steps}}$) | 0.0625 |
| Backup length | 16 |
| Burn-in | 16 |
| Batch size | 16 |
| $|\mathcal{Z}|$ | 16 |
| $\epsilon_{\text{WTA}}$ | Linearly annealed from 1 to 0 in the first 500000 steps |
| $\tau$ (target EMA) | 0.995 |
| $\beta_{\text{prior}}$ | 0.025 |
| $\beta_{\text{rec}}$ | 1 |
| $\beta_{\text{KL}}$ | 20 |

## B.4 NETWORK ARCHITECTURE

Figure 6 shows the network architecture used in the experiments. In the scaling experiments, we only multiply the width of the convolutional layers in the ResNet by the multiplier, with the sizes of other layers fixed. Table 4 summarizes the number of parameters in each component.

We use Layer Normalization (Ba et al., 2016) before the nonlinear activations in the MLP heads and after the LSTM. In addition, we apply RMS normalization (Zhang & Sennrich, 2019) to the image embeddings (after the linear layer) such that the SPR objective (cosine similarity) becomes equivalent to L2 distance between the embeddings.

Similar to DreamerV3 (Hafner et al., 2023), we use block diagonal LSTM (Van Keirsbilck et al., 2019) (with 16 blocks) to reduce the computational complexity and number of parameters.

The neural network implementation is based on `PyTorch` (Paszke, 2019), except for the block diagonal LSTM, where we used an efficient implementation from `flashrnn` (Pöppel et al., 2024).

Table 4: Number of parameters in each component.

| Component | Parameters (millions) |
|---|---|
| IMPALACNN (m=1/2/4/8) | 2 / 4 / 9 / 22 |
| LSTM | 3 |
| Transition Model | 21 |
| Value heads ($\hat{A}, \hat{B}, \hat{V}$) | 4 |

## B.5 DREAMERV3 BASELINE

We use the official reimplementation from `https://github.com/danijar/dreamerv3/`. By default, DreamerV3 uses a slightly different environment configuration compared to the Dopamine baseline, namely, higher observation resolution, full action sets, and a larger discount

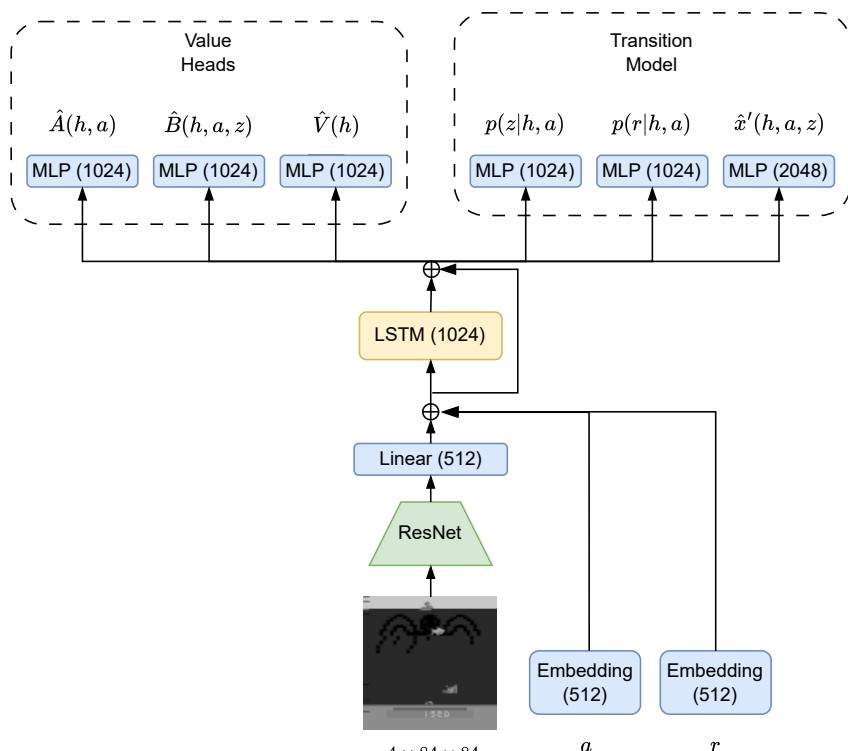

Figure 6: The network architecture. We use the same ResNet encoder proposed by Espeholt et al. (2018). All MLP heads have 1 hidden layer. Previous actions and rewards are first embedded into 512-dimensional vectors before summed together with the image embedding to form the final embedding vector. We use a residual connection around the LSTM similar to Kim et al. (2017).

factor. For a closer comparison, we lower the resolution to $80{\times}80$[6], use the minimal action sets, and lower the discount factor to 0.99.

In addition, as DreamerV3 uses a lower replay ratio by default[7], we increase its replay ratio such that the number of gradients is the same as DAE. It should be noted that DreamerV3 trains the actor-critic network and the dynamics model separately, where the actor-critic learns from trajectories generated from the learned dynamics model. This makes a direct comparison difficult, as the number of frames seen by the dynamics model and the actor-critic model differs by a factor of the rollout (imagination) length. For simplicity, we adjust the batch size such that the ground truth frames seen by the agent throughout training remains fixed.

B.6    ADDITIONAL RESULTS

Per-environment learning curves and final evaluation scores of the scaling experiments can be found in Figure 11 and Table 5.

**Correlation between HNS and prior entropy**    Here, we examine how the changes in prior entropy ($\Delta_H = H_{\text{frame-stack}} - H_{\text{LSTM}}$) relate to the relative changes in human-normalized scores

---

[6]The preset network only accepts resolutions divisible by 16, so we lower it to $80{\times}80$ instead of the standard $84{\times}84$.

[7]The definition of replay ratio in Dreamer ($\frac{\text{frames per batch}}{\text{frames per gradient}}$) is slightly different from the convention ($\frac{\text{gradients}}{\text{env. steps}}$)

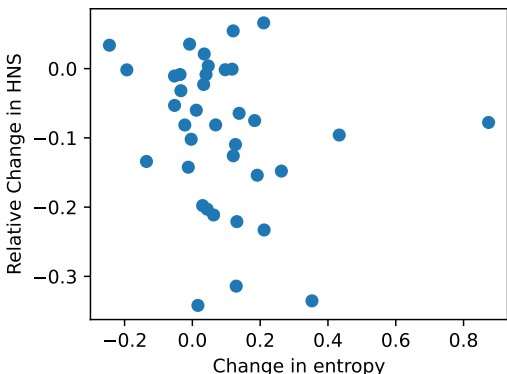

Figure 7: Scatter plot of changes in prior entropy ($\Delta_H$) and changes in HNS ($\Delta_{\text{HNS}}$) (aggregated over 5 seeds). Each point represents an environment. We remove the outliers (top/bottom 10% changes in HNS) for better visualization.

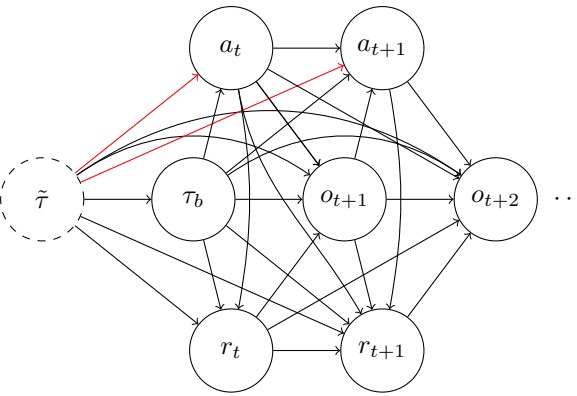

Figure 8: Causal relationship between variables of a truncated sequence for a general POMDP. $\tilde{\tau}$ denotes the truncated part of the sequence, and $\tau_b$ denotes the burn-in part of the sequence. The red arrows show the dependencies between actions and $\tilde{\tau}$ when using recurrent actors.

($\Delta_{\text{HNS}} = \frac{\text{HNS}_{\text{frame-stack}} - \text{HNS}_{\text{LSTM}}}{\text{HNS}_{\text{LSTM}}}$, we use relative changes in HNS because the scales vary considerably across environments) by comparing the LSTM agent and the Frame-stacking agent. The prior entropy is estimated by averaging the entropy of $p(\cdot|h_t, a_t)$ over training samples throughout training. We find a weak but negative (Spearman) correlation ($\rho = -0.198$), suggesting that increase in partial observability (increase in entropy) is related to decrease in performance; however, the sample size is relatively small (47 environments), and we defer a more comprehensive analysis to future work.

**Confounding** Confounding is a phenomenon in which unobserved variables influence both the actions and the outcomes, creating spurious correlations (Pearl, 2009). In the case of recurrent agents, this can happen when the behavior policy has access to variables that are not present during training.

For POMDPs, since states are replaced by histories, we have to process sequences of observations instead of singular states. In deep RL, this is typically achieved using recurrent neural networks (RNNs), such as LSTMs or GRUs (Hochreiter & Schmidhuber, 1997; Hausknecht & Stone, 2015; Mnih et al., 2016; Kapturowski et al., 2018; Gruslys et al., 2018; Cho et al., 2014; Hafner et al., 2023), but this can be computationally expensive during training when trajectories extend to thousands of steps. Instead, it is common to truncate histories by sampling random segments of contiguous trajectories from the replay buffer, and use the first few steps as the context (burn-in) before

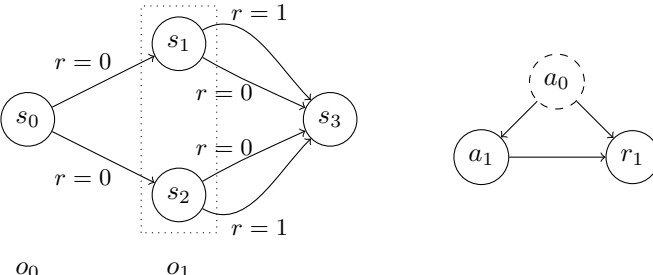

Figure 9: **Left**: A toy POMDP with 4 states and 2 actions. The nodes and the arrows represent the states and the actions (up, down), respectively. $s_0$ is the starting state and $s_3$ is the terminal (absorbing) state. The agent does not observe the underlying state but only the emitted observation at each time step, $o_0$ and $o_1$, where both $s_1$ and $s_2$ emit the same observation $o_1$. **Right**: The (simplified) causal relationship between $a_0$, $a_1$, and $r_1$. We ignore other variables as they do not influence $r_1$. The variable $a_0$ can act as a confounder during training when the target policy is memoryless.

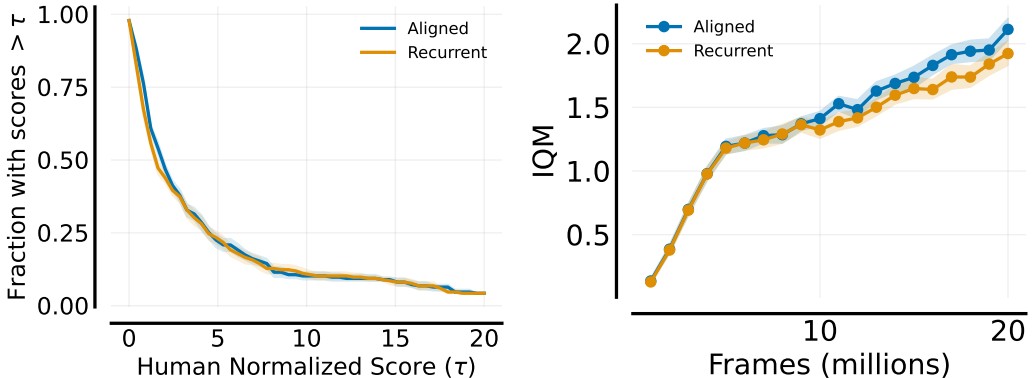

Figure 10: Effect of confounding.

updating the values (Kapturowski et al., 2018):

$$\underbrace{o_0, a_0, r_0, \cdots, o_{t-k-1},}_{\tilde{\tau}\text{ (truncated)}} \underbrace{a_{t-b-1}, r_{t-b-1}, o_{t-b}, \cdots,}_{\tau_b\text{ (burn-in)}} \underbrace{a_{t-1}, r_{t-1}, o_t, a_t, r_t, \cdots, o_{t+k}}_{\text{value updates}}$$

In this setting, the truncated part of a trajectory can act as confounders and create spurious correlation between the sampled segments and the future rewards (see Figure 8 for the causal graph). This can be mitigated by storing recurrent states in the replay buffer (Kapturowski et al., 2018); however, they may not always be available (e.g., offline settings, or policies with non-recurrent sequence models such as transformers (Vaswani et al., 2017; Parisotto et al., 2020)). If we learn the value functions (i.e., predict $\sum_{t'>t} r_{t'}$) by conditioning on $(\tau_b, a_t, r_t, o_{t+1}, \cdots)$, then $\tilde{\tau}$ can influence both the input variables and the output variables and lead to confounding.

In Figure 9, we construct a toy POMDP to illustrate this effect. In this environment, the optimal policy is $\pi^*(\text{up}|o_0) = p \in [0, 1]$ (arbitrary), and $\pi^*(a|o_0, a_0=a, o_1) = 1$ (repeat previous actions). Consider the case where the behavior policy is the optimal policy $\pi^*(\text{up}|o_0) = 0.5$, but the burn-in length is 0 (i.e., memoryless) for the target policy. In this case, we will incorrectly infer that $V^\pi(o_1) = Q^\pi(o_1, \cdot) = 1$ for any target policy $\pi$, since all the collected trajectories receive a reward 1 irrespective of the action $a_1$.

Here, we examine the effect of this misalignment between the behavior policy and the target policy in a larger scale using the ALE. Specifically, we consider two sampling strategies that differ in their dependencies on the truncated part of the trajectories (red arrows in Figure 8):

1. Aligned: the behavior policy only conditions on the past $k$ frames, where $k$ is equal to the burn-in length during training.

2. Recurrent: the behavior policy conditions on the full history.

In addition, we reduce the burn-in length (16→4), and do not store recurrent states in the replay buffer to enhance the effect of partial observability. Figure 10 shows that this subtle change in the behavior policy leads to an approximately 10% change in the IQM, suggesting that confounding should not be ignored when designing POMDP agents.

Finally, we note that, in general, deleting the red arrows is not enough to eliminate confounding since $\tau_b$ and $r_t$ can still be influenced by $\tilde{\tau}$; however, our results suggest that this simple change can already have non-trivial effects on the agent's performance.

### B.7 COMPUTE RESOURCES

All experiments were conducted using an internal cluster of Nvidia H100 GPUs. Runtime varies across environments and scales approximately proportionally with the model size ($m$), where a single run of the $m = 8$ model takes approximately 18 hours, while a run with $m = 4$ takes approximately 10 hours.

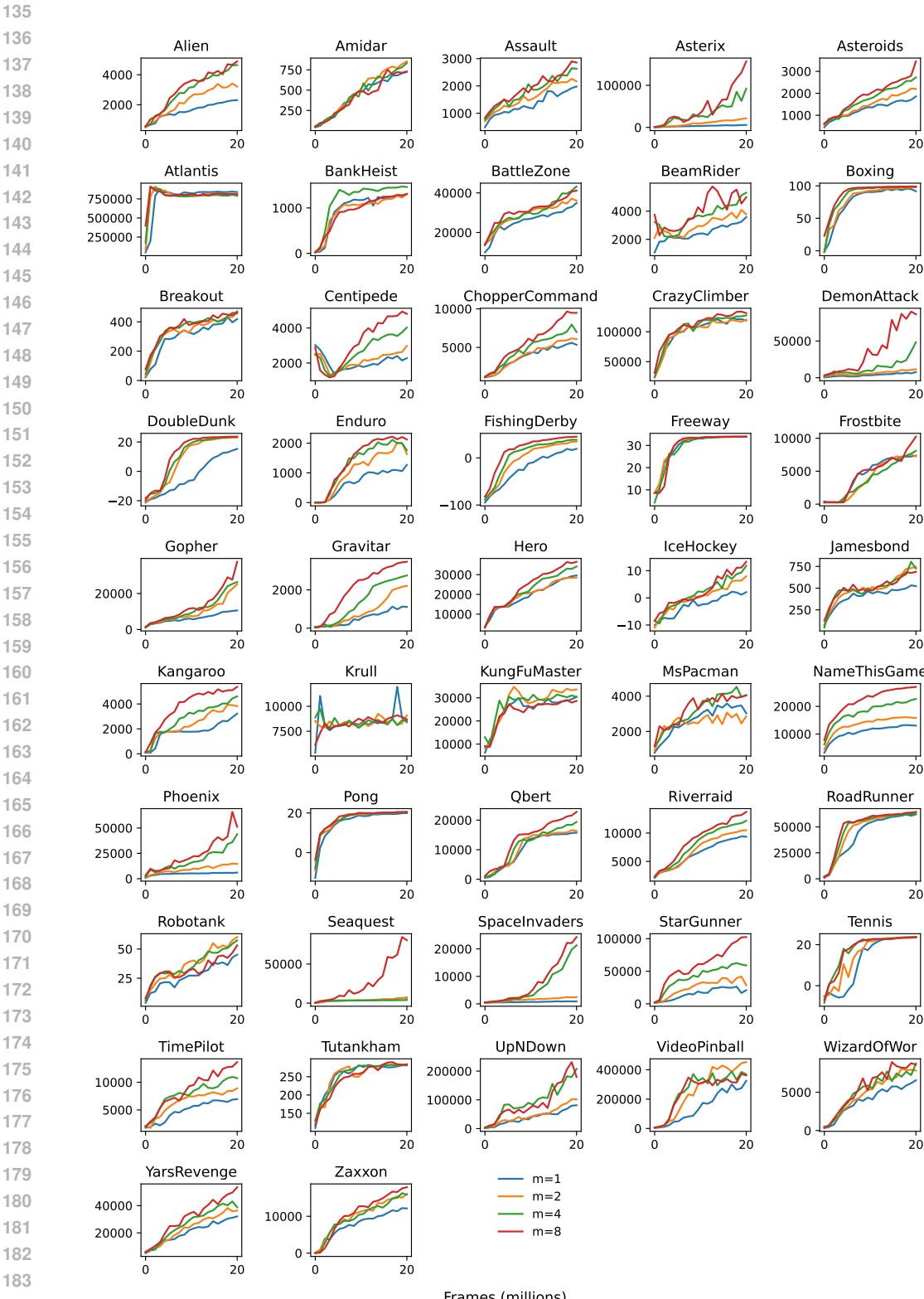

Figure 11: Per-environment learning curve (averaged over 5 random seeds).

Table 5: Per-environment final score (averaged over 5 random seeds).

| Environment | $m = 1$ | $m = 2$ | $m = 4$ | $m = 8$ |
|---|---|---|---|---|
| Alien | 2310.68 | 3211.60 | 4644.16 | 4893.08 |
| Amidar | 723.46 | 853.87 | 832.83 | 730.36 |
| Assault | 1978.36 | 2166.20 | 2628.49 | 2857.14 |
| Asterix | 5873.40 | 21777.00 | 92247.00 | 156815.40 |
| Asteroids | 1857.80 | 2198.12 | 2726.20 | 3454.60 |
| Atlantis | 839273.20 | 808648.40 | 791305.60 | 811127.60 |
| BankHeist | 1305.44 | 1304.12 | 1460.76 | 1309.12 |
| BattleZone | 34892.00 | 36060.00 | 43248.00 | 41152.00 |
| BeamRider | 3577.28 | 3772.19 | 5309.55 | 4995.91 |
| Boxing | 91.74 | 97.66 | 98.74 | 98.98 |
| Breakout | 418.86 | 461.58 | 458.20 | 470.13 |
| Centipede | 2277.04 | 2982.91 | 4028.48 | 4810.40 |
| ChopperCommand | 5342.00 | 6079.20 | 6994.40 | 9482.00 |
| CrazyClimber | 118894.80 | 120037.60 | 127332.40 | 131008.40 |
| DemonAttack | 7789.28 | 11520.46 | 48341.62 | 86428.92 |
| DoubleDunk | 15.30 | 23.06 | 23.53 | 23.65 |
| Enduro | 1269.05 | 1639.45 | 1763.89 | 2122.55 |
| FishingDerby | 19.19 | 34.46 | 38.50 | 45.09 |
| Freeway | 33.96 | 33.95 | 33.95 | 33.95 |
| Frostbite | 7360.76 | 7254.04 | 8085.24 | 10241.76 |
| Gopher | 10577.68 | 25171.12 | 26382.40 | 37633.68 |
| Gravitar | 1112.80 | 2211.20 | 2752.20 | 3469.40 |
| Hero | 29598.34 | 28531.42 | 34141.38 | 36544.68 |
| IceHockey | 2.11 | 7.94 | 11.81 | 13.30 |
| Jamesbond | 523.00 | 745.00 | 725.80 | 687.80 |
| Kangaroo | 3221.20 | 3821.60 | 4606.80 | 5374.80 |
| Krull | 8406.16 | 9092.92 | 8547.84 | 8676.52 |
| KungFuMaster | 30375.20 | 33618.40 | 30416.80 | 28644.80 |
| MsPacman | 3028.52 | 2861.52 | 4041.28 | 4056.84 |
| NameThisGame | 13042.96 | 15804.12 | 22694.48 | 27069.20 |
| Phoenix | 6029.16 | 14570.20 | 43868.00 | 51149.60 |
| Pong | 19.92 | 20.40 | 20.42 | 20.54 |
| Qbert | 15747.40 | 16345.50 | 19361.20 | 22770.80 |
| Riverraid | 9367.56 | 10520.20 | 12180.84 | 13728.08 |
| RoadRunner | 62068.40 | 63964.80 | 62404.00 | 64405.20 |
| Robotank | 45.18 | 60.28 | 57.39 | 53.23 |
| Seaquest | 7098.88 | 6303.80 | 4040.72 | 80438.08 |
| SpaceInvaders | 888.48 | 2440.16 | 21300.54 | 24328.60 |
| StarGunner | 20946.40 | 28576.40 | 58930.00 | 102573.60 |
| Tennis | 23.58 | 23.65 | 23.72 | 23.64 |
| TimePilot | 6943.60 | 8904.00 | 10742.80 | 13643.60 |
| Tutankham | 284.19 | 280.69 | 281.97 | 282.20 |
| UpNDown | 81297.36 | 101890.80 | 207808.56 | 179719.76 |
| VideoPinball | 324508.52 | 452230.68 | 369343.43 | 361586.25 |
| WizardOfWor | 6562.80 | 7830.80 | 8750.40 | 8589.60 |
| YarsRevenge | 32066.58 | 36369.90 | 38757.06 | 53512.85 |
| Zaxxon | 12138.00 | 16053.20 | 15867.60 | 17897.60 |

