# OpenReview forum: "Direct Advantage Estimation for Scalable and Sample-efficient Deep Reinforcement Learning"
_ICLR.cc/2026/Conference — ICLR 2026 Conference Withdrawn Submission_

### Official Review · Reviewer_2ngK · 2025-10-31

**Soundness:** 3
**Presentation:** 3
**Contribution:** 2
**Rating:** 4
**Confidence:** 3

**Summary:**

This paper addresses two limitations of the Off-policy Direct Advantage Estimation (DAE) method: (1) its restriction to fully observable MDPs and (2) its high computational cost, which stems from the need to learn a high-dimensional generative model of transition probabilities. The authors make two primary contributions:

1) Theoretical Extension to POMDPs: They generalize the DAE return decomposition to POMDPs. This is achieved by reformulating the problem in terms of information vectors (histories, $h_t$).
2) To address the high computational cost of modeling $p(r,o′∣h,a)$, the authors propose a novel discrete latent dynamics model.

**Strengths:**

The paper's primary contribution seems to be the discrete latent dynamics model instead of reliance on a full generative CVAE to model the transition probabilities. By combining a self-predictive objective (similar to SPR ) with a Winner-Takes-All (WTA) loss, the authors create a model that can predict a discrete distribution over next embeddings. This captures some stochasticity and bypasses the need for a high-dimensional decoder. This allows the dynamics model and the RL agent to be trained jointly end-to-end with minimal overhead , effectively solving the computational problem.

**Weaknesses:**

The first weakness of the method seems to be complexity. There are many moving parts. The final agent has many new, interacting components: a WTA annealing schedule, a delayed onset for the B correction, an adaptive temperature T for the target policy, and multiple loss coefficients. How sensitive is the final performance to this new set of hyperparameters? The stability seems to hinge on the careful annealing and balancing of the various losses.

The second weakness of the method is that the extension to POMDPs seems elementary to me. It essentially replaces the state with the history.

**Questions:**

How important are the specific architectural choices you made? How do they compare to previous works like [1-3]

[1] Mastering Atari with Discrete World Models
[2] Transformers are Sample-Efficient World Models
[3] Towards General-Purpose Model-Free Reinforcement Learning
[4] TD-MPC2: Scalable, Robust World Models for Continuous Control

See weaknesses as well.

---

### Official Review · Reviewer_gx9f · 2025-11-01

**Soundness:** 2
**Presentation:** 3
**Contribution:** 2
**Rating:** 4
**Confidence:** 3

**Summary:**

The authors present a reinforcement learning algorithm for partially observable Markov decision processes. They do so by decomposing the advantage function into two components, one of which must satisfy a centering property. The constrained term is then approximated by a conditional VAE, which the authors learn efficiently by considering a loss function that operates purely in the embedding space.

**Strengths:**

The paper considers an important and challenging problem.

The paper is well written and well organized. The key ideas are communicated in a clear fashion.

The authors provide a thorough ablation study with interesting discussions and observations.

The related work section covers relevant topics.

**Weaknesses:**

I personally find Figure 1 somewhat difficult to understand. Although conditioning on x and r would understandably clog the figure, it would potentially also improve readability.

The benchmark considered by the authors does not include continuous control benchmarks.

The proposed approach is relatively complex to understand and implement.

The authors do not discuss the potential limitations of the proposed method.

**Questions:**

How sensitive is the constraint to out-of-distribution data? is this a problem when the training is very heavily off-policy?

Line 044: limite applicability

Line 080: What is \hat{V}? Is it the same as \hat{V}_target?

Line 107-108: Can the authors provide  a reference to the statement that the tuple “is the unique minimizer of this objective function”?

The direct advantage estimation framework is somewhat complex. Would it be possible to formulate a similar approach with a more conventional advantage estimation approach?

How does the policy perform against a baseline that has access to the full state space (full observability)?

---

### Official Review · Reviewer_H9x2 · 2025-11-01

**Soundness:** 1
**Presentation:** 2
**Contribution:** 1
**Rating:** 2
**Confidence:** 4

**Summary:**

This paper extends Direct Advantage Estimation (DAE) to partially observable domains and introduces discrete latent dynamics models to approximate transition probabilities efficiently. The submission conducts experiments in the Arcade Learning Environment and shows that their approach performance is comparable to Rainbow DQN.

**Strengths:**

Please see below.

**Weaknesses:**

The theoretical contribution of the paper is a direct implication from prior work. There is no theoretical contribution provided by the submission. This is not a ground for rejection. However, how it is presented in the paper is misleading in the sense that it is claimed to have a theoretical contribution in the main body of the paper. In particular, the first bullet point is about their theoretical contribution. The alleged theoretical contributions of the submission is directly implied by existing work.

One of my major concerns with the submission is the comparison benchmark. The submission provides comparison against Rainbow DQN and claims that their proposed algorithm is comparable. In particular,
the entire comparison benchmark is just against Rainbow 200million with IMPALA CNN, Rainbow 200 million with CNN and Dreamer V3 at 20 million.  However, Rainbow is already known to be not sample efficient and there is a significant amount of publications with many proposed algorithms that perform substantially better than Rainbow.

The empirical results are based on comparisons of the submission’s proposed algorithm against algorithms that are known to be not sample efficient. Hence, the empirical analysis does not verify what is claimed in the submission.

I think the submission could benefit from thoroughly thinking about their contributions to the field and gathering proper comparisons to verify and demonstrate the claims made in their paper. This is currently not present in the paper.

Results in Figure 11 and Table 5 do not include standard deviations.

The submission also combines SPR with the Winner-Takes-All (WTA) loss. However, even SPR was not included in the comparison benchmark. Could the simple SPR be better than the submission’s proposal? We simply do not know.

Another problem with the submission is that the objective in reinforcement learning research is not boosting the performance at whatever cost or change and then claim that the proposed package of changes actually support the evidence that the proposed algorithm performs better compared to prior work. The comparisons should be fair and logical and all the confounding factors need to be isolated to understand the other changes made are not the main factor that contributes to the performance increase.

I will provide a couple examples regarding this and the submission. But throughout the submission this is a critical problem. When the submission compares their method they use a 15-layer deep residual network instead of a three-layer convolutional network and then they also provide comparison to Rainbow which has a three-layer convolutional network. This is not a useful result or information. Furthermore, when they compare to DreamerV3 it is stated that they set DreamerV3 to a 50M parameter network, but the proposed method of the submission network is 22M parameters. It is obvious that larger networks will require more samples to train. Hence, comparing these algorithms in this way does not provide any insights or evidence to the proposed method of the submission’s performance or sample efficiency compared to prior methods.

I think the paper can have the potential to be a valuable contribution. However, in its current form the submission is not ready to be published.

**Questions:**

Please see above.

---

### Official Review · Reviewer_mRMv · 2025-11-03

**Soundness:** 3
**Presentation:** 3
**Contribution:** 3
**Rating:** 6
**Confidence:** 2

**Summary:**

The paper's extension of DAE to POMDPs is theoretically sound and addresses a significant gap in scalable RL, supported by the decomposed return formulation. The latent dynamics model effectively reduces computational overhead while maintaining performance, evidenced by robustness to latent space size |Z| and entropy analysis . Various experiments performed on ALE show the effectiveness of the proposed methods.

**Strengths:**

- Theoretical generalization to POMDPs
- Comprehensive experiment settings & detailed analysis
- Clear writing makes the methodology easy to undertstand

**Weaknesses:**

- Lacking Runtime/FLOP comparisons between the proposed model and the original off-policy DAE’s generative architecture.
- Lack of compared baselines. More POMDP-based RL algorithms should be compared [1] to illustrate the effectiveness of GAE.
- The Atari benchmark is quite old, and the tasks within have been mainly solved by RL, as I know. Can the proposed DAE method be employed in robotics benchmarks, like LIBERO[2] and maniskill[3]?

[1] Recurrent Experience Replay in Distributed Reinforcement Learning. ICLR 2019

[2] LIBERO: Benchmarking Knowledge Transfer for Lifelong Robot Learning. NeurIPS 2023

[3] Demonstrating GPU Parallelized Robot Simulation and Rendering for Generalizable Embodied AI with ManiSkill3. arXiv 2024

**Questions:**

See weakness above.

---

### Note · Authors · 2025-11-15

I have read and agree with the venue's withdrawal policy on behalf of myself and my co-authors.